# Evidence of Homeostatic Regulation in *Mycobacterium avium* Subspecies *paratuberculosis* as an Adaptive Response to Copper Stress

**DOI:** 10.3390/microorganisms11040898

**Published:** 2023-03-30

**Authors:** Carlos Tejeda, Pamela Steuer, Marcela Villegas, Fernando Ulloa, José M. Hernández-Agudelo, Miguel Salgado

**Affiliations:** 1Instituto de Medicina Preventiva Veterinaria, Facultad de Ciencias Veterinarias, Universidad Austral de Chile, Valdivia 5090000, Chile; 2Escuela de Graduados, Facultad de Ciencias Veterinarias, Universidad Austral de Chile, Valdivia 5090000, Chile

**Keywords:** tolerance, MAP, copper, homeostasis

## Abstract

Background: Bacteria are capable of responding to various stressors, something which has been essential for their adaptation, evolution, and colonization of a wide range of environments. Of the many stressors affecting bacteria, we can highlight heavy metals, and amongst these, copper stands out for its great antibacterial capacity. Using *Mycobacterium tuberculosis* (*Mtb*) as a model, the action of proteins involved in copper homeostasis has been put forward as an explanation for the tolerance or adaptive response of this mycobacteria to the toxic action of copper. Therefore, the aim of this study was to confirm the presence and evaluate the expression of genes involved in copper homeostasis at the transcriptional level after challenging *Mycobacterium avium* subsp. *paratuberculoisis* (MAP) with copper ions. Methodology: Buffer inoculated with MAP was treated with two stressors, the presence of copper homeostasis genes was confirmed by bioinformatics and genomic analysis, and the response of these genes to the stressors was evaluated by gene expression analysis, using qPCR and the comparative ΔΔCt method. Results: Through bioinformatics and genomic analysis, we found that copper homeostasis genes were present in the MAP genome and were overexpressed when treated with copper ions, which was not the case with H_2_O_2_ treatment. Conclusion: These results suggest that genes in MAP that code for proteins involved in copper homeostasis trigger an adaptive response to copper ions.

## 1. Introduction

Bacteria have evolved impressive responses to a variety of stressors, which have been essential for their adaptation, evolution, and colonization in a wide range of environments [1]. These responses include processes of proteolysis and transcriptional remodeling mediated by various factors, such as some sigma types and proteases [2]. These processes can trigger changes in cell motility, cell adhesion, and protein translation [3,4]. Examples of stressors in nature are changes in the concentration of nutrients, temperature, and pH, and contact with heavy metals [5]. Among heavy metals, copper stands out for its great antibacterial capacity [6]. To reduce the negative effect of copper on the viability of bacteria, they have developed the ability to keep the intracellular concentration of this metal under control. It has been shown that the expression of bacterial copper tolerance genes generally increases in the presence of high concentrations of this metal. An example of this is the overexpression of the *cueR* gene in *E. coli* in the presence of copper [7]. In addition, it has been shown that *Mycobacterium tuberculosis* (*Mtb*) has mechanisms of copper homeostasis and tolerance associated with the action of efflux proteins (mctB and ctpV), metallothionein (mymT), transcription factors (csoR), and multicopper oxidases (mmcO) [8,9,10].

Another mycobacteria is *Mycobacterium avium* subsp. *paratuberculosis* (MAP), the causal agent of paratuberculosis, a major inflammatory bowel disease that affects ruminants [11]. MAP is one of the most fastidious members of the *Mycobacterium* genus and belongs to the *Mycobacterium avium* complex (MAC). It is a Gram-positive bacterium that is acid-fast due to its thick cell wall rich in complex lipids. It has an extremely low metabolic activity and tends to form clumps or “clusters” of bacterial cells. In addition, it can form biofilms. These biological features make MAP highly resistant to adverse environmental conditions [11,12]. A novel treatment based on copper ions was shown to be highly effective in reducing MAP load in liquid matrices, such as PBS and milk [13,14]. Despite this great efficacy, a small proportion of MAP cells managed to tolerate the treatment [14,15].

These findings are consistent with the known ability of MAP to withstand unfavorable environments [16], as well as a wide variety of physical and chemical stressors usually harmful to the vast majority of mesophilic bacteria [17]. The adaptation of MAP in stressful environments has been explained by the expression of genes linked to its ability to withstand the intracellular environment of macrophages [18,19]. The synthesis of serine proteases in response to acidic pH environments has also been detailed [20,21]. However, there is no published information regarding copper as a MAP stressor and the adaptive response of the pathogen to this metal.

The fact that some level of tolerance to copper ion treatment has been described justifies the question of whether MAP has similar homeostatic mechanisms to those described in other bacteria to regulate copper at the intracellular level. For this reason, the objective of this study was to confirm the presence and explore the adaptation mechanisms of MAP to copper through the evaluation of the expression of genes involved in copper homeostasis.

## 2. Material and Methods

### 2.1. Selection of Genes Involved in Copper Homeostasis

In order to generate information related to copper homeostasis in MAP, we set out to determine the presence of genes in MAP which are potentially linked to copper homeostasis. To do this, we took certain genes linked to copper homeostasis in *Mtb* as a reference [9,10,22].

We attempted to detect the presence of the following 5 genes in MAP: *csoR*, which encodes a transcription repressor factor; *ctpV*, which encodes an influx protein; *mctB*, which encodes a membrane channel efflux protein; *mmcO*, which encodes a multicopper oxidase protein; and *mymT*, which encodes a metallothionein protein.

First, a bioinformatics analysis was performed to ascertain the presence of these genes in MAP in silico. The complete genome of MAP strains DSM 44135 (access number: CP053068.1) and K10 (access number: A3016958.1) [23,24] were used in which homologous genes (i.e., those whose sequences are similar) to those described in *Mtb* strain H37Rv (accession number: CP003248.2) were searched for, using the BlastN search tool (http://www.ncbi.nlm.nih.gov/BLAST/ (accessed on 7 June 2022)).

#### Primer Design and Confirmation of Gene Presence

Once the homologous genes in MAP had been identified, the nucleotide sequences were used to design specific primers for each gene, using the Blast-Primer tool (https://www.ncbi.nlm.nih.gov/tools/primer-blast/ (accessed on 7 June 2022)) [25]. Subsequently, and with the aim of confirming the presence of the genes and optimizing the PCR reaction, a gradient PCR was performed using the previously designed primers, and the MAP ATCC 19698 strain was used to obtain the DNA template. A 25 μL reaction volume was used, containing 12.5 μL GoTaq^®^ Green Master Mix, 2X (Promega, Madison, WI, USA), 1 μL of each primer at a concentration of 10 μM, 5.5 μL of nuclease-free water, and 5 μL of DNA from each treatment. An Axigen^®^ Maxygene II thermal cycler was used with the following program for the gradient PCR: initial denaturation step at 94 °C for 5 min, followed by 40 cycles (denaturation at 94 °C for 1 min, annealing [51 °C–53 °C–55 °C–57 °C–61 °C–63 °C] for 1 min, and elongation at 72 °C for 1 min), and a final elongation step at 72 °C for 10 min.

Finally, the fragments obtained were visualized through a 1% agarose gel using SYBR Safe (Invitrogen^TM^, Waltham, MA, USA), a highly sensitive nucleic acid staining reagent widely applied in molecular biology.

### 2.2. Exploring the Adaptive Responses of MAP to Copper Stresses

#### 2.2.1. Experimental Design

In the second stage of this study, through an in vitro experiment under controlled conditions, gene expression was evaluated after a copper ion treatment. The study design considered a volume of 500 mL of PBS inoculated with MAP at a concentration of 10^6^ cells/mL. It comprised two types of challenges with stressors, plus a non-challenged control group. The experimental groups were (1) suspension of PBS + MAP treated with copper ions for 15 min (450 ppm); (2) suspension of PBS + MAP treated with H_2_O_2_ for 15 min (oxidative control not related to the functionality of copper homeostasis genes); and (3) suspension of PBS + MAP not treated with any stressors.

#### 2.2.2. Preparation of MAP Cultures

As a study model to determine the presence of the target genes, MAP ATCC 19698 strain was used, which was cultivated in a 7H9 liquid medium supplemented with 10% oleic acid–albumin–dextrose–catalase (OADC) (Becton Dickinson and Company, Sparks, MD, USA), 2 mg/L mycobactin J (Allied Monitor, Fayette, MO, USA), and 2 mL/L glycerol for 1 month at 37 °C [26]. MAP growth was monitored weekly using a UV/VIS UV-9200/VIS-7220G spectrophotometer (Rayleigh). When the absorbance at 600 nm reached a value of 1.0, it was estimated to be in late exponential growth at a concentration of ~10^8^ MAP cells mL^−1^ with minimal dead cells present [26]. Then, 10-fold serial dilutions of MAP were made in PBS, and the 10^−6^ dilution was used for the experiments. All bacterial suspensions were kept at 4 °C for no longer than 24 h until use.

#### 2.2.3. MAP Detection and Quantification

To confirm MAP presence in each experimental unit and in the control once challenged with copper ions, first, DNA was extracted from MAP cells suspended in the buffer. Then, it was quantified using a NanoDrop 2000 spectrophotometer (Thermo Scientific, Vilnius, Lithuania). Subsequently, the DNA was subjected to a qPCR protocol based on the *IS900* sequence detection in a QuantStudio™ 3 system (Thermo Scientific, Vilnius, Lithuania). The qPCR reaction consisted of 5 μL DNA template (600 ng), 10 μL of 2x TaqMan Universal Master Mix (Roche, Mannheim, Germany), 0.2 μM primers, and 0.1 μM probe, making up a final volume of 20 μL. The sequences of the primers used to amplify a 63-nucleotide fragment of the *IS900* gene were 5′-gacgcgatgatcgaggag-3′ (L) and 5′-gggcatgctcaggatgat-3′ (R). The amplification conditions were as follows: One cycle at 95 °C for 10 min; 45 cycles with three steps of 95 °C for 10 s, 60 °C for 30 s, and 72 °C for 1 sec; and a final cooling step at 40 °C for 30 s. A blank sample (only PCR water) and positive (MAP ATCC 19698) control were used for both the DNA extraction protocol and the PCR reaction [15]. MAP cell counts were estimated according to the genome equivalent principle [13] based on the concentration of MAP DNA that was measured in the same spectrophotometer mentioned above. This concentration was adjusted for a 10^8^ dilution and to the number of copies of the *IS900* target gene, having the reference of the molecular weight of the genome of MAP ATCC strain 19698 to establish a standard curve for estimation of MAP numbers in the sample by a QuantStudio™ 3 system (Thermo Scientific, Vilnius, Lithuania) real-time PCR, according to the following equation:$$Genome\;equivalent=\frac{DNA\;concentration\;\left(\mathrm{ng}/\mu L\right)\times \left(6.022\times 10^{23}{\mathrm{mol}}^{-1}\right)}{\begin{array}{c} \left(4.829\times 10^{6}base\;pairs\right)\times \left(1\times 10^{9}\mathrm{ng}/g\times 660g/\mathrm{mol}\right)\\ \left(MAP\;ATCC\;19698\;genome\right)\;\;\;\;\;\;\;\;\;\;\;\left(Base\;mass\right) \end{array}}$$

#### 2.2.4. Treatment with Copper Ions

The buffer contained in the pure copper device was subjected to an electrical stimulus according to the protocol described by Steuer et al. [13]. This consisted of a glass beaker containing 500 mL of PBS inoculated with MAP into which 2 high-purity copper plates were introduced. These plates were stimulated with a low voltage (24 V) electrical current (3 amps) to allow the release of copper ions. The time of exposure was modified to 15 min. A magnetic stirrer placed into the beaker allowed constant homogenization during the treatment.

#### 2.2.5. Determination of Copper Concentration

To determine the total copper concentration in the PBS buffer after being challenged, each of the aforementioned treatments was digested with concentrated HCl (37% *w/w*) and HNO_3_ (0.1 N) according to a modified protocol [15]. The measure was then determined by an atomic absorption spectrophotometer (AAS), and the result was reported in ppm. This measurement determination was run in triplicate.

#### 2.2.6. Analysis of Gene Expression

To assess whether there were any changes in the expression of those genes to copper homeostasis in the MAP strain under study, a qPCR protocol was used to estimate relative expression [8,27]. To accomplish this, RNA was isolated from each stressor-treated and untreated suspension using the TRIzol method, as recommended by the manufacturer (Invitrogen™, Waltham, MA, USA). Briefly, 250 μL aliquot of each sample was mixed with 750 μL of TRIzol reagent, incubated for 5 min, and then 0.2 mL of chloroform was added. The samples were centrifuged for 15 min at 12,000× *g* at 4 °C. The aqueous phase was then transferred to another tube, and the RNA was precipitated by adding 0.5 mL of isopropanol. The samples were centrifuged at 12,000× *g* at 4 °C for 10 min, and the resulting supernatant was discarded. RNA was washed by adding 1 mL of 75% ethanol, mixing the sample, and centrifuging at 7500× *g* at 4 °C for 5 min. The supernatant was discarded, and the tubes were left open for 5 min in order to dry the pellet. Then, 35 μL of nuclease-free water was added to resuspend the pellet and kept at −80 °C until use. Then, the RNA was quantified, and its purity was evaluated using the Thermo Scientific™ NanoDrop 2000 kit. Following this, complementary DNA (cDNA) was synthesized using M-MLV Reverse Transcriptase (Promega, Madison, WI, USA), according to the manufacturer’s instructions. Briefly, an aliquot of the extracted RNA was mixed with the primer adaptor (Oligo(dT)15 Primer (Promega, Madison, WI, USA)) and nuclease-free water in a total volume of 15 μL. The RNA aliquot was calculated to obtain 2 μg of total RNA, and the primer-adaptor volume was calculated to obtain 0.5 μg per each μg of RNA. Each tube was heated at 70 °C for 5 min and cooled immediately on ice. Then, to each tube, 5 μL of M-MLV 5X reaction buffer (Promega, Madison, WI, USA), 1.25 μL of dNTPS mix (Promega, Madison, WI, USA), 0.63 μL of recombinant RNasin^®^ ribonuclease inhibitor (Promega, Madison, WI, USA), 1 μL of M-MLV RT (Promega, Madison, WI, USA), and 2.12 μL of nuclease-free water were added to a final volume of 25 μL. Then, the samples were incubated for 60 min at 42 °C and kept at −20 °C until use. Finally, a qPCR protocol was performed using SYBR green (Agilent Brilliant II) [22], the synthesized cDNA, the primers designed for this study, and the QuantStudio™ 3 thermocycler (ThermoFisher). The program consisted of an initial denaturation step at 95 °C for 10 min, followed by 40 cycles, and a final elongation step at 72 °C for 45 s. The cycles consisted of denaturation at 95 °C for 45 s, annealing (57 °C for the primers of potential copper homeostasis genes, and 60 °C for the constitutive gene (*gapDH*) [28]) for 45 s). The melting process consisted of heating to 95 °C for 15 s, 60 °C for 1 min, and 95 °C for a further 15 s. Nuclease-free water was used as blank. Subsequently, the cycle threshold (Ct) values of the cDNA samples were normalized in relation to those of the housekeeping gene *gapDH*.

Finally, these threshold values were used to calculate the relative expression by the comparative method 2^−ΔΔCt^ [27,29], according to the formula:2^−ΔΔCt^ = 2 ^−[(Ct of the gene of interest in the treated sample − Ct of the housekeeping gene in the treated sample) − (Ct of the gene of interest in the untreated sample − Ct of the housekeeping gene in the untreated sample)]^(1)

This method makes it possible to estimate the quantification of gene expression for each copper homeostasis gene in samples treated with copper ions in relation to those not treated.

#### 2.2.7. Statistical Analysis

Because the data were not normally distributed, to estimate the difference in gene expression between copper ion treatment and treatment with the oxidative stressor H_2_O_2_, the nonparametric Wilcoxon rank sum test was used. Analyses were performed using the R program version 3.1.2 (R Development Core Team 2015) with a *p* value < 0.05.

## 3. Results

We were able to identify five MAP homologous genes that could be involved in Cu homeostasis: *csoR*, *ctpV*, *mctB*, *mmcO*, and *mymT* based on previous studies carried out on *Mtb*. Specific primers were designed for each gene, and through gradient PCR, the ideal annealing temperature for the 5 genes was estimated as 57 °C (Table 1). The sequence similarity between copper homeostasis genes in MAP and those in *Mtb*, estimated by the Blast tool, were: *ctpV* (69.39%), *csoR* (96%), *mctB* (77.46%), *mmcO* (80.71%), and *mymT* (83.33%).

The presence of *Mtb* homologous genes associated with copper homeostasis in the MAP genome was confirmed, and their respective amplicon sizes were visualized in agarose gel: *csoR* (140 bp); *ctpV* (81 bp); *mctB* (157 bp); *mmcO* (140 bp); *mymT* (103 bp) (Figure 1).

All genes showed an increase in expression (2^−ΔΔCt^) when treated with copper ions in contrast to those treated with H_2_O_2_, as estimated by qPCR (*csoR* gene: 34.37, *ctpV* gene: 210.43, *mctB* gene: 136.59, *mmcO* gene: 39.31, *mymT* gene: 50.19 with copper ions versus *csoR* gene: 1.28, *ctpV* gene: 1.28, *mctB* gene: 1.81, *mmcO* gene: 2.13, *mymT* gene: 1.04 treated with H_2_O_2_), with the genes that code for efflux proteins (*ctpV* and *mctB*) being those that showed the greatest difference in expression compared to those treated with H_2_O_2_ (Figure 2).

In addition, the median value of copper homeostasis gene expression was significantly higher (*p* < 0.05) in MAP cells subjected to copper ion treatment than it was in H_2_O_2_-stressed MAP cells (Figure 3).

## 4. Discussion

The results of this study show the overexpression of five genes associated with copper homeostasis once the MAP cell has been challenged with this metal.

These results are consistent with studies that have shown that these genes have functionality in various bacterial models, as they are highly conserved among various groups of bacteria [30]. For example, there is published information that shows evidence of homeostatic mechanisms in bacterial cells, such as the reference strain K-12 of *Escherichia coli* or *Enterococcus hirae*, which, when stressed with copper, increase the expression of genes that encode proteins with different functions, such as Cu-specific chaperones and storage and efflux systems [30,31,32].

The high degree of homology between these MAP genes and those described in *Mtb* (due to their close sequence similarity) can be considered a proof of concept of the aforementioned assumption, given the conserved nature of this type of gene. In addition to this, taking *Mtb* as a model was also justified from an evolutionary standpoint since there is a close relationship between certain mycobacteria and related bacteria, which share families of proteins for specialized secretion systems, such as secretion type VII, which is partially involved in the virulence of some mycobacteria (e.g., ESX-1) and particular families of mycobacterial proteins such as PE/PPE [33]. Although the complete sequencing of the MAP genome [23,24] allowed us to gain information regarding the adaptive response of this pathogen to different environmental conditions such as pH and temperature [5], there was no information about the adaptive response of MAP to the copper stressor.

Besides confirming the presence of this type of gene, we also set out to understand how MAP generates an adaptive response to the copper stressor, i.e., to see the functionality of these genes. To do this, we used the differential expression of certain genes [34,35]. It is known that bacteria can modify their gene expression in response to environmental conditions, such as temperature changes, which is a result of epigenetics associated with DNA methylation, which likewise produces changes in their gene expression when exposed to the stressor [36].

What is known about the functionality of the proteins encoded by some of these five genes under study and how it is related to viability–tolerance of MAP in the presence of copper is that, e.g., the *csoR* gene and its encoded protein are stimulated in the presence of high copper concentrations, as is the expression of other genes involved in copper homeostasis, such as the *ctpV* gene, all of which are regulated by the *cso* operon [37].

The role played by *ctpV* in the pathogenicity of mycobacteria has been demonstrated, since animals infected with a strain of *Mtb* in which part of the *ctpV* gene had been mutated manifested greater damage of the lung lobes and a more severe granulomatous response [22].

It seems that MAP also has the ability to mount an adaptive response to a copper stressor, which would explain the observed tolerance to these stressors in some MAP cells [38,39]. Regarding the effect of copper on MAP, Tejeda et al. [39] found that the cell wall of MAP did not suffer any type of alteration once treated with copper ions. It is known that in the cell wall of MAP and other mycobacteria [10], efflux proteins are present. The increased expression of the *mctB* gene, which encodes an efflux pump protein essential to tolerating and maintaining low levels of Cu at the intracellular level [9], and which is reported in the present study, could then be an argument that explains the copper tolerance observed in some MAP cells. The *mctB* gene is related not only to tolerance to metals, but also to antibiotics [9].

It was observed that exposure of MAP to H_2_O_2_ had no effect on the expression of copper homeostasis genes. This oxidizing agent was used as a negative control for those genes that modulate copper homeostasis. Tejeda et al. [39] reported that this reagent exerted a low level of stress on MAP viability compared to the stress caused by copper ions. These reactive oxygen species (ROS) do not have an effect on viability by themselves, but it has been shown that, together with other ROS, they affect the integrity of some bacteria [40], including MAP [41,42].

Finally, it would be interesting to propose a study that considers the tolerance–virulence relationship, as evidence has been shown that bacterial strains that are more tolerant to any stressor agent are also more virulent. Mehtar et al. [43], for example, studied the response of some strains of *Mtb*, which were exposed to copper. Some of them tolerated this treatment and were the ones that showed more virulence in experimental animals. Wolschendorf et al. [9] also demonstrated that copper tolerance is associated with pathogen virulence. Steuer et al. [14,44], on the other hand, demonstrated that although copper ions are highly effective in reducing the number of MAP in milk intended for calves, some MAP cells survived this copper-based treatment and were associated with greater evidence of pathology in these calves. Therefore, more studies should explore whether copper treatments are selecting more virulent MAP strains as described with many antibiotics [45]. If this potential MAP virulence ability is a product of its tolerance driven by an adaptative homeostatic response to copper, this information should alert us to consider improvements in these treatments that are also capable of giving rise to possible tolerant strains.

## 5. Conclusions

The presence and expression of genes that are related to copper homeostasis are confirmed in the MAP genome. This finding suggests a possible adaptive response by the bacterium to copper ion stress.

## Figures and Tables

**Figure 1 microorganisms-11-00898-f001:**
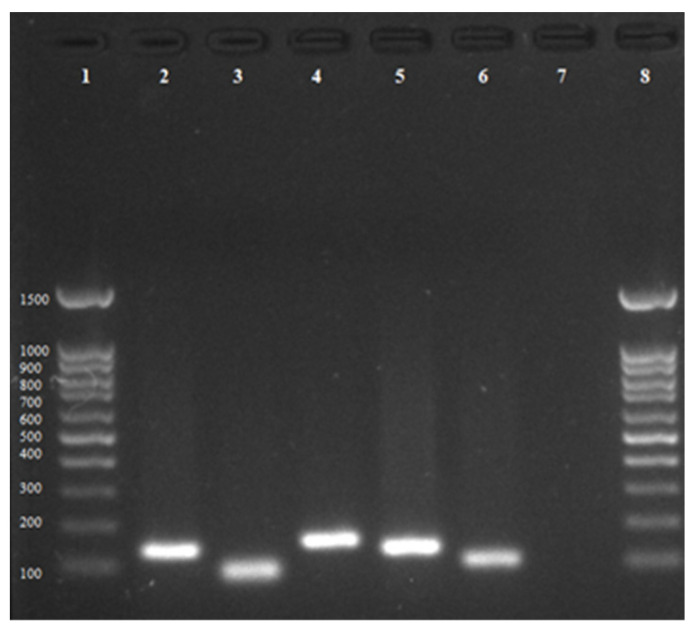
Genes related to copper homeostasis in MAP. Lane 1: 100 bp standard; lane 2: *csoR* gene; lane 3: *ctpV* gene; lane 4: *mctB* gene; lane 5: *mmcO* gene; lane 6: *mymT* gene; lane 7: negative control; lane 8: 100 bp standard.

**Figure 2 microorganisms-11-00898-f002:**
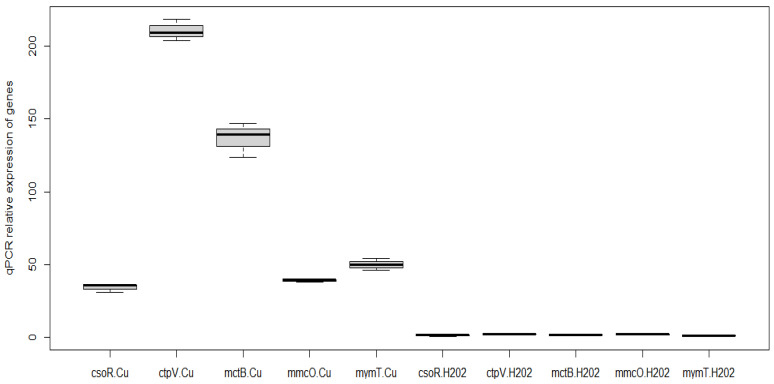
Relative expression of the copper homeostasis genes measured by qPCR, evaluated in samples treated with copper ions and H_2_O_2_. The data were analyzed with the ΔΔCt method, using *gapDH* as a control gene, and the mean Ct values of the copper homeostasis genes in each treated and untreated sample.

**Figure 3 microorganisms-11-00898-f003:**
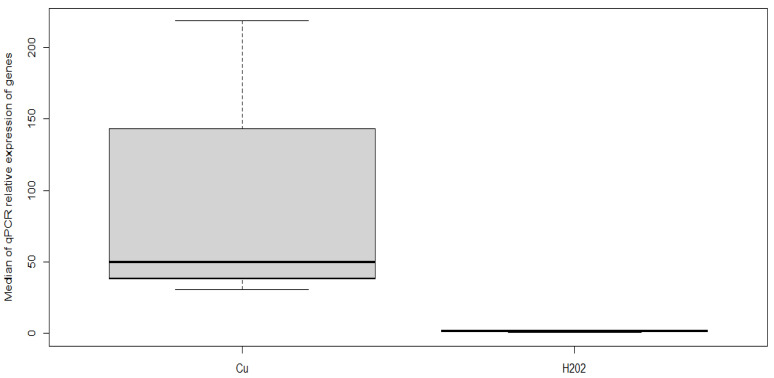
Comparison of the median value of expression of copper homeostasis genes evaluated by qPCR in samples treated with copper ions and H_2_O_2_. The data were analyzed with the ΔΔCt method, using *gapDH* as a control gene, and the median Ct values of the copper homeostasis genes in each treated and untreated sample.

**Table 1 microorganisms-11-00898-t001:** Gene identification, primers sequences, and annealing temperature for each gene used in this study.

Gene	Primer Sequence (5′-3′)	Annealing Temperature (Tm)
*csoR*	GCGCTGATCTGAGTGAGGA	57 °C
	ATGACCAACGAACACGGGTA	
*ctpV*	CCAACCTCAAACATGGCGTC	57 °C
	CGTACAGCGACCACAGGAAG	
*mctB*	TGATCTCGCTACGCCAACAC	57 °C
	TCGTTGAGCCCGTTGATCTG	
*mmcO*	GGCGGCAACATGATCCAGTA	57 °C
	TGCAGGTGAATCGGGTGATAC	
*mymT*	CGGAACCCTGCTGACCTG	57 °C
	ACAGGTGCAGCGGTACTC	
*gapDH*	ATCGGGCGCAACTTCTACC	60 °C
	GTCGAATTTCAGCAGGTGAGC	

## Data Availability

Not applicable.

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
