# Peer review of "Evidence of Homeostatic Regulation in *Mycobacterium avium* Subspecies *paratuberculosis* as an Adaptive Response to Copper Stress"

_microorganisms, 2023, doi:10.3390/microorganisms11040898_

Round 1

Reviewer 1 Report

This is an interesting manuscript on adaptative responses of MAP to copper stress. The manuscript is interesting but some aspects need to be revised.

1-    Introduction.

Line 44- stands that the ctpV gene of M. tuberculosis encodes for an influx protein related to copper homeostasis. On the contrary, that protein has been described to have an efflux function, as it is showed in the Reference 22 of this manuscript (Ward, S. K et al. Molecular microbiology. 2010, 77(5), 1096-1110). Please, correct that point.

2-    Methods.

2.2.3.

* The DNA was isolated from the different conditions tested and then quantified.

Please clarify how that quantification was made, the current text mentioned two different spectrophotometers used for quantification of DNA (line 127 and line 139).

* Moreover, to perform qPCR 5 microliters of DNA was used. Could authors indicate the amount of DNA (e.g. nanograms) present in that volume?

* To perform “genome equivalent” calculations, authors consider the genome size of the strain ATCC 19698 as reference (lines141-142). Unfortunately, that genome has not been sequenced yet. How authors calculated the genome size of that strain?

* What’s the aim of amplifying IS900? Please clarify.

2.2.6.

* To determine the responses of MAP to copper and hydrogen peroxide, authors collected the bacteria from a PBS suspension. Line 112 indicates that MAP is also collected from PBS without any stressor. It is unclear that RNA isolated from bacteria without stressors does represent the “untreated samples”, Could authors explain that point in lines 197-199?

3-    Results:

* Sequence of primers used for the housekeeping gene (gapDH) should be also indicated in Table 1.

* Why the gene gapDH was used as housekeeping? Have authors checked if gapDH gene behaves as real housekeeping in the conditions applied: does it amplify in similar levels along the conditions tested?

4-    Discussion:

* line 272. Modify the sentence as follow “….how it is related to viability-tolerance of MAP in the presence of copper, …”

5-    Conclussion:

* This paragraph should be corrected.

There are many full genomes known of MAP (see NCBI data base) therefore, the presence of the five genes studied in the MAP genome is already known; moreover, data are not indicating a function for the five genes under study, they just indicate that their expression level was associated to copper.

Authors should talk about gene expression suggestive of an adaptative response to copper. Please modify the sentence of conclusions accordingly.

Author Response

This is an interesting manuscript on adaptative responses of MAP to copper stress. The manuscript is interesting, but some aspects need to be revised.

1-    Introduction.

Line 44- stands that the ctpV gene of M. tuberculosis encodes for an influx protein related to copper homeostasis. On the contrary, that protein has been described to have an efflux function, as it is showed in the Reference 22 of this manuscript (Ward, S. K et al. Molecular microbiology. 2010, 77(5), 1096-1110). Please, correct that point.

Author´s response (AR): the reviewer is correct. The ctpV gene codes for a cytoplasmatic membrane efflux protein. The sentence has been corrected as suggested. Page (P) 1, Line (L) 44, new version (NV).

2-    Methods.

2.2.3.

* The DNA was isolated from the different conditions tested and then quantified.

Please clarify how that quantification was made, the current text mentioned two different spectrophotometers used for quantification of DNA (line 127 and line 139).

AR: the reviewer is correct. The sentence has been corrected. P3, L 140-141 NV

* Moreover, to perform qPCR 5 microliters of DNA was used. Could authors indicate the amount of DNA (e.g. nanograms) present in that volume?

AR: the amount of DNA used was 600 nanograms in the 5 microliters. This information has added in the text, as suggested. P3, L 131, NV.

* To perform “genome equivalent” calculations, authors consider the genome size of the strain ATCC 19698 as reference (lines141-142). Unfortunately, that genome has not been sequenced yet. How authors calculated the genome size of that strain?

AR: the information regarding MAP ATCC 19698 genome size estimation was published by Wu et al. 2009. DOI:  10.1186/1471-2164-10-25

* What’s the aim of amplifying IS900? Please clarify.

AR: the IS900 element has been extensively used as a specific target for MAP identification-confirmation. This insertion element has multiple copies (17-20) which allow a higher sensitivity for a PCR system than when it is achieved with single-copy targets (Stabel et al., 2004; Herthnek and Bölske, 2006; Irenge et al., 2009).

2.2.6.

* To determine the responses of MAP to copper and hydrogen peroxide, authors collected the bacteria from a PBS suspension. Line 112 indicates that MAP is also collected from PBS without any stressor. It is unclear that RNA isolated from bacteria without stressors does represent the “untreated samples”, Could authors explain that point in lines 197-199?

 AR: the “untreated samples” are the samples taken before the copper-treatment started, when copper concentration was 0 ppm, i.e., time 0 samples also called control samples.

These control samples (samples not challenged with copper) were crucial to estimate copper effect on gen expression. The data obtained when the equation of ΔΔCt was applied, should be interpreted as the expression of the gene of interest relative to the internal control in the treated samples (e.g., those exposed to copper ions) compared with the untreated samples (those not exposed to copper ions).

3-    Results:

* Sequence of primers used for the housekeeping gene (gapDH) should be also indicated in Table 1.

 AR: the sequence of the gapDH gene has been added in table 1, as suggested.

* Why the gene gapDH was used as housekeeping? Have authors checked if gapDH gene behaves as real housekeeping in the conditions applied: does it amplify in similar levels along the conditions tested?

 AR: Glyceraldehyde3-phosphate dehydrogenase (GapDH) is a housekeeping gene commonly used to standardize data in real-time PCR experiments (Rokbi et al. 2001; Granger y col. 2004). In the present study, this gene was similarly expressed under different conditions tested (treated and untreated samples), showing very similar Ct values.

4-    Discussion:

* line 272. Modify the sentence as follow “…. how it is related to viability-tolerance of MAP in the presence of copper, …”

AR: the sentence has been changed, as suggested. P7, L 273-274, NV.

5-    Conclusion:

* This paragraph should be corrected.

There are many full genomes known of MAP (see NCBI data base) therefore, the presence of the five genes studied in the MAP genome is already known; moreover, data are not indicating a function for the five genes under study, they just indicate that their expression level was associated to copper.

Authors should talk about gene expression suggestive of an adaptative response to copper. Please modify the sentence of conclusions accordingly.

AR:  in general terms the reviewer is correct, but the confirmation of the presence of these five genes in MAP genome, hasn’t been previously studied (it has been only described in Mycobacterium tuberculosis). In fact, the primers used were design by our research team. Although, the functionality of these genes was not part of the aim of the study, the conclusion was modified, as suggested. P8, L 314, NV.

Reviewer 2 Report

Map is a microorganism with very specific characteristics and high resistance to environmental conditions. In my opinion, the paper lacks an explanation why this type of mucobacterium was chosen for the experiment.

Author Response

Map is a microorganism with very specific characteristics and high resistance to environmental conditions. In my opinion, the paper lacks an explanation why this type of mycobacterium was chosen for the experiment.

AR: background information that justifies the use of this important pathogen as a study model in the present study, has been extensively provided in the intro part. P2, L54-71, NV.

Reviewer 3 Report

  1. Abbreviation When used for the first time, you write the complete phrase and the abbreviation between brackets. Ex. MAP lines 19 and 20.
  2. Lines 19 and 20 methodology: In line 19, buffer contaminated with MAP was treated with two stressors (or left untreated), replace "contaminated" with "inoculated."
  3. Line 25: H2O2 rewrite H2O2
  4. The authors must add reference to each primer in table 1 and transfer to materials and methods section.
  5. Figure 1: unclear must be improved; Lane 1: 100 bp standard How? and must be estimated pb for each gene from five genes.
  6. Line Evaluation of the adaptive response of MAP to copper as a stressor rephrase.
  7. Lines 106 and 107 The study design 106 considered a volume of 500 mL of PBS contaminated with MAP, rewrite and replace the word contaminated.
  8. Line 108 delete words as an experimental unit.
  9. Lines 148 and 149 This consisted of a glass beaker containing 148,500 mL of PBS contaminated with MAPP, Rewrite and replace the word contaminated.
  10. Line 157 HNO3, should be corrected to HNO3.
  11. On lines 209 and 210 we were able to identify 5 MAP homologous genes that could be involved in Cu homeostasis rephrase this sentence.
  12. It should add a relative expression of the copper homeostasis genes of untreated with any stressor MAP to figure 2.

Author Response

  1. Abbreviation When used for the first time, you write the complete phrase and the abbreviation between brackets. Ex. MAP lines 19 and 20.

AR: the reviewer is correct. The sentence has been corrected, as suggested. P1, L 19-20, NV.

  1. Lines 19 and 20 methodology: In line 19, buffer contaminated with MAP was treated with two stressors (or left untreated), replace "contaminated" with "inoculated."

AR: the word “contaminated” has been changed by “inoculated”, as suggested. P1, L 20, NV.

  1. Line 25: H2O2 rewrite H2O2

AR: the molecular formula has been rewritten as suggested. P1, L 25, NV.

  1. The authors must add reference to each primer in table 1 and transfer to materials and methods section.

AR: we humbly disagree with the reviewer, since the primers used in this study were specifically design by our research team. Please see P2, L 88-91, NV., and P5, L 212, NV. The primer sequence of gapDH gene was the only one not designed by us, hence it was accordingly cited in the text.

  1. Figure 1: unclear must be improved; Lane 1: 100 bp standard How? and must be estimated pb for each gene from five genes.

AR: lane 1 is the known base pair standard (100 bp in this case). Normally, this standard is used in order to estimate the amplicon size of the genes of interest. The information regarding the size of each amplicon can be found in line P5, 220, NV.

  1. Line 104 Evaluation of the adaptive response of MAP to copper as a stressor rephrase.
  1. The sentence has been rephrased for clarity, as suggested. P3, L104, NV.

  1. Lines 106 and 107 The study design 106 considered a volume of 500 mL of PBS

contaminated with MAP, rewrite and replace the word contaminated.

AR: the word “contaminated” was replaced for “inoculated”. P3, L109, NV.

  1. Line 108 delete words as an experimental unit.

AR: the words were deleted as suggested. P3, L108, NV.

  1. Lines 148 and 149 This consisted of a glass beaker containing 148,500 mL of PBS contaminated with MAPP, Rewrite and replace the word contaminated.

AR: the word was replaced, as suggested. P4, L151, NV.

  1. Line 157 HNO3, should be corrected to HNO3.

AR: the molecular formula was corrected as suggested. P4, L159, NV.

  1. On lines 209 and 210 we were able to identify 5 MAP homologous genes that could be involved in Cu homeostasis rephrase this sentence.

AR: the sentence has been rephrased for clarity, as suggested. P5, L211-213, NV.

  1. It should add a relative expression of the copper homeostasis genes of untreated with any stressor MAP to figure 2.

AR: I'm afraid it is not possible, because the ΔΔCt value is interpreted as the expression of the gene of interest relative to the internal control in the treated sample compared with the untreated sample, therefore it is implicit in the formula and you have just one “fold change” value and not one before treatment and another after treatment.

Reviewer 4 Report

This manuscript had studied the gene expression of homeostatic regulation genes in Mycobacterium avium subspecies paratuberculosis under treatment by copper and concluded homeostatic regulation serves as an adaptive response to copper stress. The observation is interesting, however, it lacks further insightful elucidation. Thus, my suggestion is rejection.

Firstly, and most importantly, this manuscript contains very limited data, and the authors had conducted copper treatment and q-PCR on a few genes relevant to homeostatic regulation, no more and no less. It looks more to me that this is the first observation of a good story/study. However, why not dig deep to answer the next questions step by step, and thus drive the study to go further. For example, what causes the alteration in the expression of homeostatic regulation genes, and how did the authors reckon only such few genes but not other genes play an important role? This may require some global expression analysis like RNAseq or multiple omics. How does copper concentration influence such regulation, and how to further confirm this in other mimicking models (why in the first place we need to study copper treatment)? At last, genetic manipulation to confirm the role of homeostatic regulation is also required.

Secondly, the English writing needs to be improved.

Author Response

This manuscript had studied the gene expression of homeostatic regulation genes in Mycobacterium avium subspecies paratuberculosis under treatment by copper and concluded homeostatic regulation serves as an adaptive response to copper stress. The observation is interesting, however, it lacks further insightful elucidation. Thus, my suggestion is rejection.

Firstly, and most importantly, this manuscript contains very limited data, and the authors had conducted copper treatment and q-PCR on a few genes relevant to homeostatic regulation, no more and no less. It looks more to me that this is the first observation of a good story/study. However, why not dig deep to answer the next questions step by step, and thus drive the study to go further. For example, what causes the alteration in the expression of homeostatic regulation genes, and how did the authors reckon only such few genes but no other genes play an important role? This may require some global expression analysis like RNAseq or multiple omics. How does copper concentration influence such regulation, and how to further confirm this in other mimicking models (why in the first place we need to study copper treatment)? At last, genetic manipulation to confirm the role of homeostatic regulation is also required.

AR: we naturally disagree with the reviewer. As the reviewer himself establishes, this study represents a first approach to try to explain how MAP could be adapted to the stressful effect of copper. Similar approaches were proposed by Ward et al., 2010 and Wolschendorf et al. 2011. Shi et al., 2015, working with Mycobacterium tuberculosis as a model. Given the high level of resistance and tolerance that characterize this pathogen, it was highly recommended to propose this study design first, and once we have established some degree of adaptive response, we can then get into in all the aspects mentioned by the reviewer y futures studies.

Secondly, the English writing needs to be improved.

AR: the manuscript was English proof-read, before submission, by Paul Dissington (paulmfsingi@hotmail.com), who is a native English speaker from Liverpool (UK), who holds the academic degree of Bachelor of Education in English

Round 2

Reviewer 3 Report

Thank you for revising the manuscript according to the suggestions.

Author Response

we thank the reviewer's opinion.

Reviewer 4 Report

The manuscript, entitled “Evidence of homeostatic regulation in Mycobacterium avium subspecies paratuberculosis as an adaptive response to copper stress” described that copper homeostasis genes were present in the MAP genome and were overexpressed when treated with copper ions, which was not the case with H2O2 treatment. According to the comments I given in last round of review, the authors had revised the manuscript, however, due to some significant drawbacks, my suggestion is rejection.

Comments:

1. First and foremost, this study shows too few results. The only data is RT-PCR. The authors can also supplement the data on the variation of transcription level with the time of cultivation or the transcription level changes of five genes with the concentration of copper ions.

2. In the Figure 2, why did not the authors show the data on the control group which was not treated with any stress?

3. In Abstract section, “(or left untreated)” should be deleted, because the authors did not mention the figures in this sentence.

4. In line 82, the first line should be indented.

5. The format of “2 -ΔΔCt” and “2-ΔΔCt” in lines 198 and 199 is not consistent,

6. In line 122, “mL1” should expressed as “mL-1

7. In section of “2.2.3. MAP detection and quantification”, the format of symbol “º C” is not correct.

8. In lines 180 and 185, both “ul” and “uL” appear in the manuscript.

Author Response

Response to Reviewer 4 comments

Comments and Suggestions for Authors

The manuscript, entitled “Evidence of homeostatic regulation in Mycobacterium avium subspecies paratuberculosis as an adaptive response to copper stress” described that copper homeostasis genes were present in the MAP genome and were overexpressed when treated with copper ions, which was not the case with H2O2 treatment. According to the comments I given in last round of review, the authors had revised the manuscript, however, due to some significant drawbacks, my suggestion is rejection.

Comments:

  1. First and foremost, this study shows too few results. The only data is RT-PCR.

Author Response (AR): we again disagree with the reviewer, since the study offers many more results than those reported by the reviewer in his last comment. For example, on page 5, line 212, the confirmation of the presence of the 5 copper homeostasis genes studied is reported, using a conventional PCR system. In addition, on the same page, line 213, genomic similarity results are provided, obtained from an in silico approach of sequence similarity using the BLAST tool. Furthermore, on page 6, lines 225-231, the differential expression of genes is reported and figures 2 and 3 show the quantification of this expression.

The authors can also supplement the data on the variation of transcription level with the time of cultivation or the transcription level changes of five genes with the concentration of copper ions.

AR: the scientific question that the present study tried to answer was to confirm the presence of copper homeostasis genes in Mycobacterium avium subsp. paratuberculosis, together with the differential expression of these genes when challenged with copper ions. Even though what the reviewer proposes has an interesting rationale, it corresponds to a different scientific question than the one originally proposed to answer with the present study.

  1. In the Figure 2, why did not the authors show the data on the control group which was not treated with any stress?

AR: the ΔΔCt value is interpreted as the expression of the gene of interest relative to the internal control in the treated sample compared with the untreated sample, therefore the data of the control group is implicit in the formula and you have just one “fold change” value and not one before treatment (i.e., for the control group) and another after treatment.

  1. In Abstract section, “(or left untreated)” should be deleted, because the authors did not mention the figures in this sentence.

AR: the sentence was modified as suggested.

  1. In line 82, the first line should be indented.

AR: the reviewer is correct; the line was corrected as suggested.

  1. The format of “2 -ΔΔCt” and “2-ΔΔCt” in lines 198 and 199 is not consistent.

AR: the reviewer is correct; the format was corrected as suggested.

  1. In line 122, “mL−1” should expressed as “mL-1

AR: the reviewer is correct; the volume unit was modified as suggested.

  1. In section of “2.2.3. MAP detection and quantification”, the format of symbol “º C” is not correct.

AR: the reviewer is correct; the symbol was corrected as suggested, page 3, lines 136-138 of the edited version (EV).

  1. In lines 180 and 185, both “ul” and “uL” appear in the manuscript

AR: the reviewer is correct; the volume unit was corrected.